# Improvements to Supercomputing Service Availability Based on Data Analysis

**Jae-Kook Lee** , **Min-Woo Kwon, Do-Sik An, Junweon Yoon** , **Taeyoung Hong, Joon Woo, Sung-Jun Kim** and **Guohua Li** *

National Supercomputing Center, Korea Institute of Science and Technology Information, 245 Daehak-ro, Yuseong-gu, Daejeon 34141, Korea; jklee@kisti.re.kr (J.-K.L.); mwkwon81@kisti.re.kr (M.-W.K.); dsan@kisti.re.kr (D.-S.A.); jwyoon@kisti.re.kr (J.Y.); tyhong@kisti.re.kr (T.H.); wjnadia@kisti.re.kr (J.W.); sjkim@kisti.re.kr (S.-J.K.)

* Correspondence: ghlee@kisti.re.kr; Tel.: +82-42-869-1689

**Abstract:** As the demand for high-performance computing (HPC) resources has increased in the field of computational science, an inevitable consideration is service availability in large cluster systems such as supercomputers. In particular, the factor that most affects availability in supercomputing services is the job scheduler utilized for allocating resources. Consequent to submitting user data through the job scheduler for data analysis, 25.6% of jobs failed because of program errors, scheduler errors, or I/O errors. Based on this analysis, we propose a K-hook method for scheduling to increase the success rate of job submissions and improve the availability of supercomputing services. By applying this method, the job-submission success rate was improved by 15% without negatively affecting users' waiting time. We also achieved a mean time between interrupts (MTBI) of 24.3 days and maintained average system availability at 97%. As this research was verified on the Nurion supercomputer in a real service environment, the value of the research is expected to be found in significant service improvements.

**Keywords:** high-performance computing; supercomputing service; data analysis; service availability; resource scheduler; resource utilization

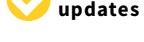



## 1. Introduction

Supercomputers are used to perform computationally intensive simulations and analyses in fields such as climate research, molecular modeling, physical simulation, cryptography, geophysical modeling, automotive and aerospace design, financial modeling, and data mining. Ensuring the availability of large cluster systems, such as supercomputers, is challenging. Many organizations also operate a supercomputer to analyze the job scheduling log data of the supercomputing users. They can then find the causes of problems and remedy them to improve service availability [1–5].

In [1], in the job-scheduling log, a large majority of jobs was reported as failures due to user behaviors such as leaving bugs in code, incorrect configurations, or operation errors. To reduce the job failure rate, similarity-based event-filtering analysis was applied, which indicated that the mean time between interrupts (MTBI) was about 3.5 days. In [2], the authors designed their own log analysis tool with a novel filtering method to effectively reduce fatal events and reached a mean time between fatal events (MTBFE) of approximately 1.3 days on their cluster system. In [3], failure categorization was performed by analyzing a five-year failure record to determine the causes of failure on users' jobs. From analyzing these papers, we found that the types of data sources used for the logs are different because the system configurations of supercomputers are complex. Although all data generated through job scheduling were analyzed, the causes of job failures also differed depending on the services provided or on queue policies. From this perspective, to increase service availability, the issues to be resolved are as follows: (1) to analyze user-submitted job data

on the causes of job failures by application category, and (2) to improve service availability under conditions that do not increase user job waiting time.

In this study, we analyzed in more detail the success and failure rates of jobs using collected user-submitted job data and examined the causes of job failure. To provide high service availability while maintaining system stability, we applied a K-hook with scheduling to increase the job-submission success rate. We describe our operational technique; present our system's MTBI, which is an indicator of system stability [6–8]; and analyze our system availability statistics.

The rest of this paper is organized as follows: Section 2 describes the system configuration, which includes the hardware and software structures of the Nurion system, and Section 3 provides the main problem statements. Section 4 introduces the proposed method, which increases the service availability. Section 5 analyzes the results of applying our method. Finally, the conclusions are presented in Section 6.

## 2. System Configuration

### 2.1. Hardware Configuration

The Nurion system is a cluster-type supercomputer [9], the hardware architecture of which is illustrated in Figure 1. It consists of various infrastructure nodes connected by a high-performance interconnect, including computing nodes, storage systems, login nodes, management nodes, datemover (DM) nodes, and web servers.

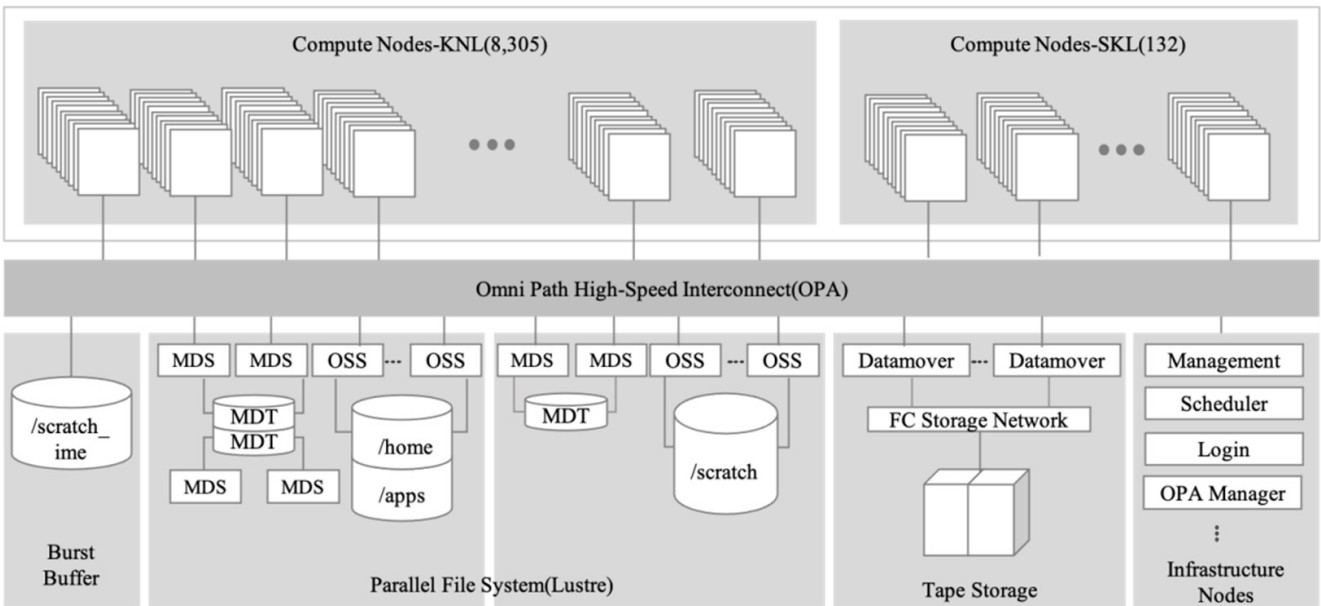

**Figure 1.** Hardware architecture of the Nurion supercomputer.

The computing nodes that perform parallel processes at high speeds consist of an 8305 Intel many-core processor Knights Landing Nodes (KNLs) [10,11] and a 132 Xeon server processor Skylake Nodes (SKLs) [12]. Each KNL node has 68 cores per socket and 96 GB (16 GB × 6) memory, while each SKL node has two sockets, each of which has 20 cores and 192 GB (16 GB × 12) memory. The interconnection network uses a 100 Gbps Intel Omni-Path Architecture (OPA) [13], and the storage uses a Lustre parallel file system [14], consisting of a user home directory (/home01), an application directory (/apps), and a user working directory for global shared scratch (/scratch) with 21 PB or more available space. To improve the speed of programs using large-scale bulk data, a burst buffer (/scratch_ime) with 800 TB or more storage capacity was deployed to provide file input/output (I/O) at a maximum speed of 800 GB/s. Tape storage with a capacity of 10 PB or more was developed to backup important system data and for long-term storage of user data. This was connected to the DM node, which handles data transfer through the

fiber channel (FC) network. The infrastructure nodes, such as the login nodes, for users of the supercomputing service and the management node for system administrator access, are connected to the Korea Research Environment Open NETwork (KREONET) [15] through a 10 Gbps Ethernet.

### 2.2. Software Configuration

Nurion is a Linux-based system that uses Bright Cluster Management (BCM) software for provisioning and managing each node. In addition, a batch scheduler (PBS Pro) is used to manage user-submitted jobs efficiently, and Intel OPA management software is used to manage the interconnection network. Lustre is used as the parallel file system; it uses DDN IME with burst-buffer software for high-performance I/O. To support the execution of various user programs, a range of compilers and message passing interface (MPI) libraries are provided, and commercial software, such as ABAQUS, ANSYS, and GAUSSIAN, can be used.

We used the scheduler's queue, which is a group of computing nodes that provide particular services depending on the user's purpose. Based on the node type, for KNL, we divided the submissions into exclusive, normal, long, flat, and debug queues. For SKL, we divided them into commercial and norm_skl queues. The job-submit policy for each queue is defined by the maximum time (wall-clock time (WCT)), the maximum number of submitted jobs, and the maximum number of running jobs per user. User-submitted jobs are executed in the order of priorities that are periodically calculated based on user jobs' wait time, requested resource size, and the predefined queue priorities in which jobs reside. All submitted jobs follow the job-submit policies in Table 1. An exclusive queue is for dedicated resources that support large projects and group research. Normal is a general resource queue for both free service users (creative research field, national strategy field, and innovation support field) and paid service users. Long is a general resource queue that requires long-term work and can be used for up to 120 h (5 days). Flat is also a general resource queue with a flat memory that can designate MCDRAM and DDR4 for up to 102 GB of memory use. Debugging is a queue for debugging KNL nodes, in which a general resource queue and interactive work can be used for debugging (shared resources). Commercial is a general resource queue for commercial applications (shared resources). Norm_skl is a general resource queue on SKL nodes, which plays the same role as the KNL normal queue.

**Table 1.** Job-submit policies by queues.

| Type | Queue | Total Nodes | Total CPU Cores | Wall Clock Limit (Hours) | Max. Submit Jobs | Max. Running Jobs |
|------|-------|-------------|------------------|---------------------------|-------------------|--------------------|
| KNL | exclusive | 2600 | 176,800 | unlimited | 100 | 100 |
| | normal | 4970 | 337,960 | 48 | 40 | 20 |
| | long | 300 | 20,400 | 120 | 20 | 10 |
| | flat | 180 | 12,240 | 48 | 20 | 10 |
| | debug | 20 | 1360 | 48 | 2 | 2 |
| SKL | commercial | 118 | 4720 | 48 | 6 | 2 |
| | norm_skl | 118 | 4720 | 48 | 10 | 5 |

From the user perspective, users can submit jobs according to this supercomputer queue policy. In this process, waiting time is an important factor for users. The users' waiting time is controlled by setting the maximum number of submitted jobs and maximum number of running jobs for each queue. An appropriate configuration for the maximum number of submitted jobs and running jobs is essential because it is an environment that provides supercomputing services to users.

### 3. Problem Statement

When a user's job fails, the scheduler notifies the user through a failure message by mail. This message contains only a simple summary; for detailed error messages, the *.log

and *.err files are referred to in the user home directory. We collected the log data from the scheduler server and output the statistics as follows.

The main factors that influence the job failure rate are summarized in Figure 2. Failure due to program errors accounted for the largest percentage, 44.4%, of the total. Next, PBS scheduler errors, such as job script errors and WCT limit errors, accounted for 37%; and I/O errors, due to incorrect file permissions, incorrect files, or directory specifications, accounted for 12.6%. Failures due to hardware errors such as memory allocation errors or system bus errors accounted for 0.6%, errors due to the user's forced termination signal accounted for 2.3%, and other unidentifiable errors accounted for 3.1%. These statistics were obtained from analyzing the error codes of job failures.

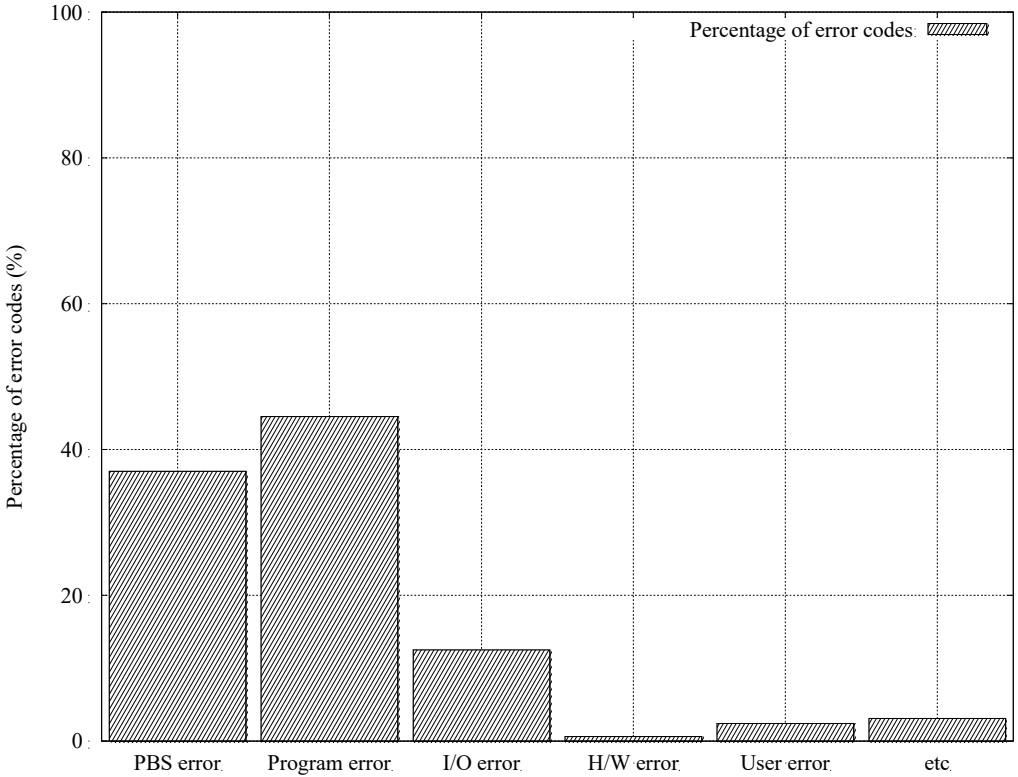

**Figure 2.** Classification of the cause of failed jobs by error codes.

A detailed analysis of each item is shown in Table 2. For PBS error, "job deletion with qdel includes walltime limit" occurred most frequently, at 36.37%. In this case, most of the jobs submitted by users either exceed the walltime limit in the queue policy or are canceled by the user arbitrarily with the qdel command. Therefore, we excluded this cause of job failure from the target to be solved. For program errors, "operation not permitted" occurred most frequently, at 41.01%. By further analyzing this case, we found that when communication with the LDAP server fails, the operation does not work normally because of I/O ownership and r/w/x permission issues of the process. For I/O errors, "no such file or directory" occurred most frequently, at 10.37%. In this case, analysis showed that the cause of the problem was a file system mount error (OPA connection). For H/W errors, "no such device or directory" occurred the most (0.31%). One cause of this problem is that MPI communication occurs between compute nodes. Another cause is that most of it overuses the memory resource. The compute nodes are of the diskless type to ensure that all computation data will be allocated on the memory. If the data are loaded into memory because of previously executed jobs, an I/O message indicating insufficient memory resources is returned. For user errors, "termination signal" occurred most frequently, at 2.05%. From this statistic, we attempted to add a prologue hook before scheduling a resource and an epilogue hook after a job is completed.

**Table 2.** Detailed description of the causes of job failures.

| Item | The Cause of Job Failure | Percentage (%) |
|---|---|---|
| PBS error | Job was requeued (if rerunnable) or deleted (if not) | 0.01 |
| | Job execution failed, do retry | 0.04 |
| | Job execution failed, before files, no retry | 0.12 |
| | **Job deletion with qdel includes walltime limit** | **36.37** |
| | Licensed CPUs exceeded | 0.09 |
| | Undefined attribute | 0.36 |
| | PBS etc. | 0.001 |
| Program error | **Operation not permitted** | **41.01** |
| | Argument list was too long | 0.04 |
| | Exec format error | 0.47 |
| | Command not found | 1.91 |
| | Stack overflow | 0.56 |
| | Segmentation fault | 0.28 |
| | Floating-point exception | 0.07 |
| | Illegal instruction | 0.1 |
| I/O error | Operation requires sequential file organization and access | 0.41 |
| | I/O procedure was truncated | 0.005 |
| | **No such file or directory** | **10.37** |
| | Input/output error | 0.2 |
| | Bad file descriptor | 1.34 |
| | Too many open files | 0.24 |
| H/W error | No such device or address | 0.31 |
| | No child processes | 0.01 |
| | Resource temporarily unavailable | 0.26 |
| | Cannot allocate memory | 0.03 |
| | Bus error | 0.01 |
| User error | Abort signal | 0.18 |
| | Kill signal | 0.09 |
| | Termination signal | 2.05 |
| etc. | etc. | 3.07 |

## 4. Method

We attempted to solve this problem by adding K-hook functions to crisis scheduling to increase the success rate of job submissions based on the analysis of errors that occur in job submissions. The overall workflow of scheduling with K-hook and a detailed flowchart are described in this section.

### 4.1. Workflow with K-Hook

We designed and implemented the workflow shown in Figure 3. Users submit jobs through the command line interface and submit job requests to the login nodes. Each request is sent to the daemon based on the server_priv configuration file and communicates with the compute nodes through the scheduler daemon and communication daemon based on the scheduling policy written in sched_priv and communication policy written in comm_priv on the PBS scheduler node. Based on the resource request transmitted through the communication daemon, the user's job is executed by exec_job on the resource-allocated compute nodes through the MoM daemon based on the mom_priv configuration file. We added K-hook to MoM running on the compute nodes rather than Server and Scheduler on PBS Pro scheduler nodes. The reason was to reduce the load of the scheduler nodes, which handle thousands of compute nodes in a huge HPC environment. As our supercomputer scheduled over 8000 compute nodes, we added K-hook to MoM on compute nodes. When a job is submitted, it is checked with the functions defined through K-hook, and if it does not pass then the node is processed offline, and the Server is notified through Communication. Server again attempts to allocate this job request to another node through Scheduler. This principle increases the success rate of job submission. The offline nodes are rebooted while attempting to recover through the self-recovery script we wrote.

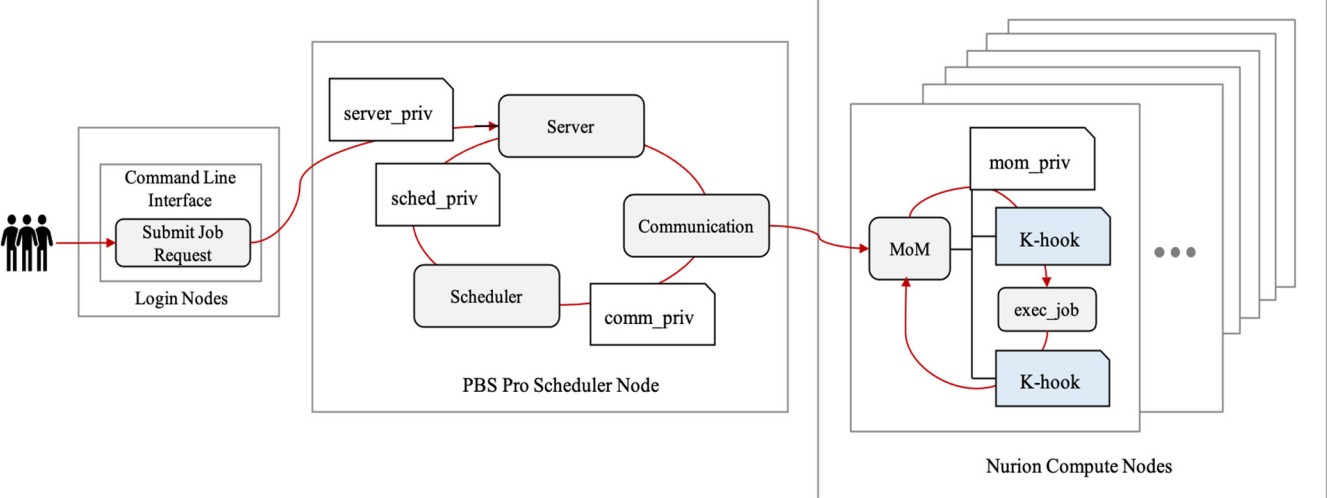

**Figure 3.** An overall workflow of scheduling with K-hook.

In this process, adding K-hooks may increase the user's waiting time. If a job fails, the user does not have to wait but must resubmit the job instead. We attempted to solve the problem of increased waiting time caused by adding a K-hook in various ways. One approach involved setting the maximum number of submitted jobs and executed jobs in the job submission policy. As our supercomputer sets different policies for each queue, we verified that waiting time does not increase with K-hooks by each queue.

K-hook uses Python plugins provided by PBSPro hook. The main functions of K-hook are implemented in a bash script, which is called through PBSPro hooks in Python. The bash script can be extensible to any other batch schedulers.

### 4.2. Flowchart with K-Hook

Scheduling flows in large-scale, high-performance computing systems are complex. In particular, resource and job scheduling for thousands of computing nodes in our supercomputing environment is even more complex. Figure 4 shows a flowchart of scheduling with K-hooks. This flowchart is based on the default settings of PBS Pro scheduling. When a job is submitted, this job will be in a queue with queue_job by Server. After Server queues the job, it is moved or modified depending on the configurations with modify_job and move_job. When a job is moved, it turns to run_job and is sent by Server to the MoM daemon in the compute nodes.

When the MoM receives a job, the user's job execution starts from this moment with execjob_begin. We applied K-hook with a memory size check function to execjob_begin. Before executing execjob_prologue, we implemented and applied three functions to execjob_begin. First, we checked the communication with the LDAP server. Communication failure causes an error in which the submitted job does not operate normally because of ownership and r/w/x permission issues. This function solves the error message of "Operation not permitted" in Program Error of classification (Table 2). Second, when the job is executed on a multi-node, communication between nodes is checked. If communication between nodes does not work properly within a limited time, in the worst case, the job process will be killed. This function solves the error message of "No such device or address" in H/W error of classification (Table 2). Third, the memory space of computing nodes is checked. The computing node in Nurion is a diskless system, and all files stored on the local node are loaded in memory. Therefore, the local file (in the/tmp directory) created by the previous user's job might run out of memory, which can cause memory errors notified as "Resource temporarily unavailable" or "Cannot allocate memory" in H/W error of classification (Table 2).

When the execution of execjob_begin with K-hook is applied and completed, MoM runs the job as a top shell and user program. If the user's program is a parallel process, such

as MPI, it is executed through tm_spawn and pbs_attach. After the job runs successfully, MoM kills the job and, through execjob_epliogue, the job is cleaned up. After this, we added K-hook functions to execjob_end to help increase the success rate.

We implemented and applied three functions to execjob_end. First, we checked the mount points of the Lustre storage. All computing nodes in Nurion mount the Lustre storage, consisting of a home directory (/home01), an application directory (/apps), and a scratch directory (/scratch), to perform a user's job. If any of these are not mounted properly, problems may occur with the jobs on these nodes. This function solves the error message, "No such file or directory", which occurred the most in I/O error classification (Table 2). Second, a check was performed for zombie processes, which can occur after a job ends. Zombie processes, which remain "dead" on computing nodes, can degrade the computational performance when the next user's job is assigned. Finally, the STREAM benchmark test was conducted to measure memory bandwidth. By applying K-hook to execjob_end, it is possible to understand comprehensively the status of the computing node and perform offline procedures when an error occurs consistently, and therefore for the administrator to check the status of the node and take appropriate action.

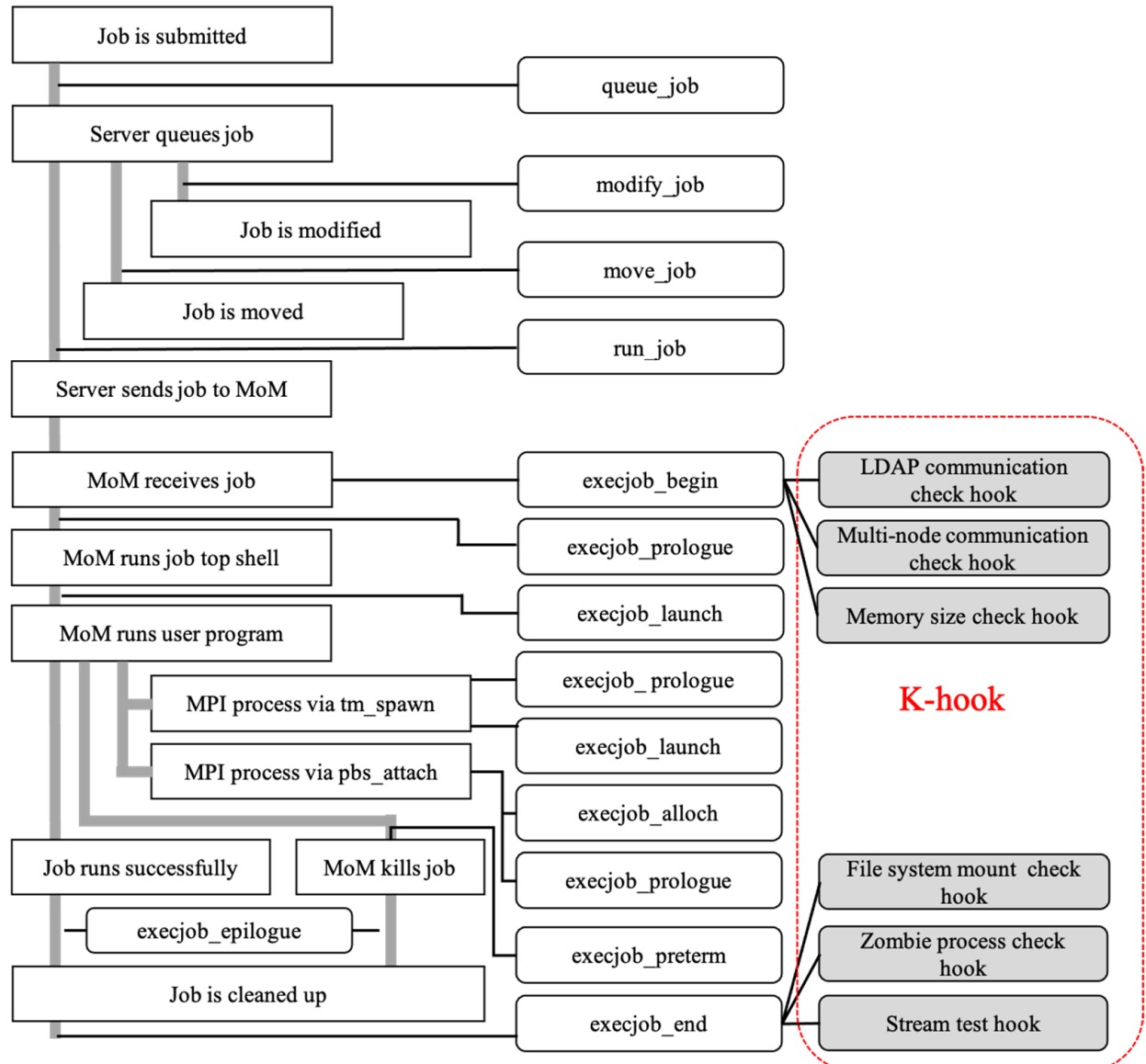

**Figure 4.** A flowchart of scheduling with K-hook.

## 5. Result

We confirmed that the success rate increased by comparing the results before and after applying K-hook to scheduling. We also confirmed whether waiting time increased by adding these functions through data analysis. In addition, the suitability of our method was verified by calculating the MBTI and system availability, which were used as indicators for the supercomputing service.

### 5.1. Evaluation of Summitted-Job Success Rate

K-hook was applied to the scheduling at the end of May 2019. We graphed the average job-submission success rate from January to May and the job-submission success rate from June to December, as shown in Figure 5. The left *y*-axis shows the number of submitted jobs, and the right *y*-axis shows the job success rate. The bar graph shows the submitted jobs count, and the line with points indicates the job-submission success rate. It can be seen that the job success rate does not affect the number of jobs. The blue line indicates the average success rate before applying K-hook, and the red line indicates the average success rate after applying K-hook. The average success rate before application was 74.4%, but the average success rate after application was 85.6%, which is an improvement of 15%. By analyzing the reason for the improved job success rate, the K-hook functions were defined based on the analysis of the existing failure error message, and the detected node information was changed to offline and transmitted to the scheduler, reducing the failure rate.

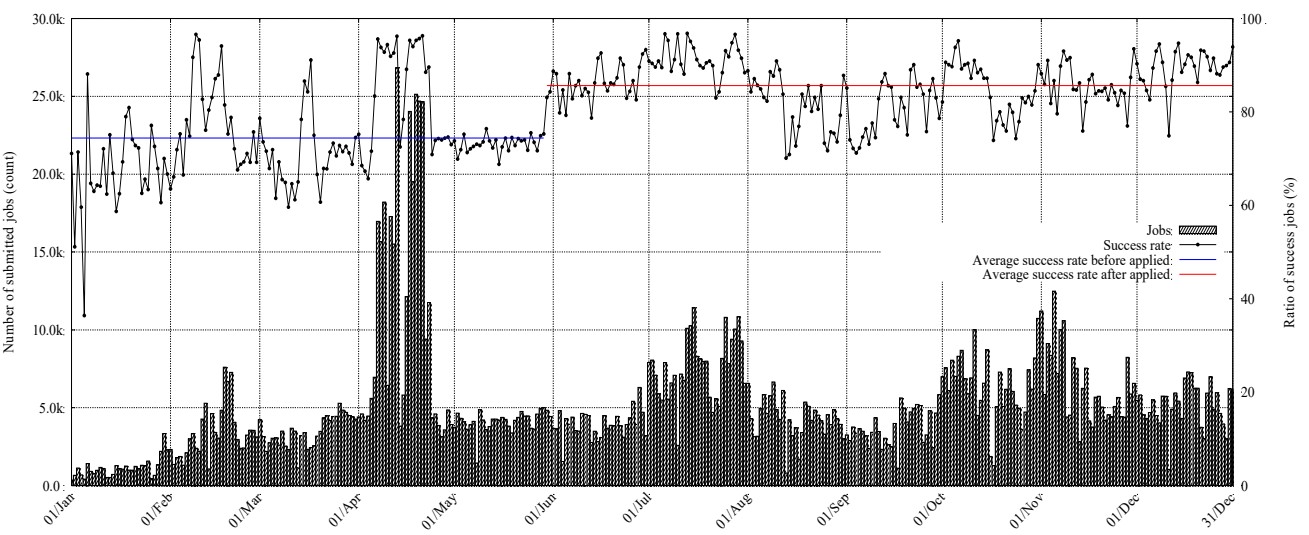

**Figure 5.** Comparison of job success rate before and after applying K-hook with scheduling.

### 5.2. Analysis of Waiting Time for Main Queues

As described in the previous section, adding K-hooks may increase users' waiting time. We modified the queue policy to reduce the waiting time of users. In terms of service availability, we reduced the maximum number of submitted jobs and executed jobs to increase job success rate without affecting waiting time. Figure 6 shows the results of analyzing the waiting time for main queues by modifying the policy at the time of adding K-hooks. The main queues to which many computing resources were allocated are the normal, exclusive, norm_skl, and commercial queues. We analyzed the waiting times of these four main queues monthly to confirm that our method does not affect waiting times. The left *y*-axis shows the number of jobs and the right *y*-axis represents the waiting time (second) by month. The bar graph shows the number of jobs, the jagged line indicates the waiting time, the blue line shows the average waiting time before application, and the red line shows the average waiting time after application. In the normal queue shown in Figure 6a, the average waiting time after application was 1376.9 s, which is 34% lower than

before K-hook was applied. In the exclusive queue shown in Figure 6b, the average waiting time after application was 14,412.1 s, a reduction of 48.9%. In the norm_skl queue shown in Figure 6c, the average waiting time after application was 1582.9 s, a reduction of 80.1% In the commercial queue shown in Figure 6d, the average waiting time after application was 5416 s, a reduction of 10.5%.

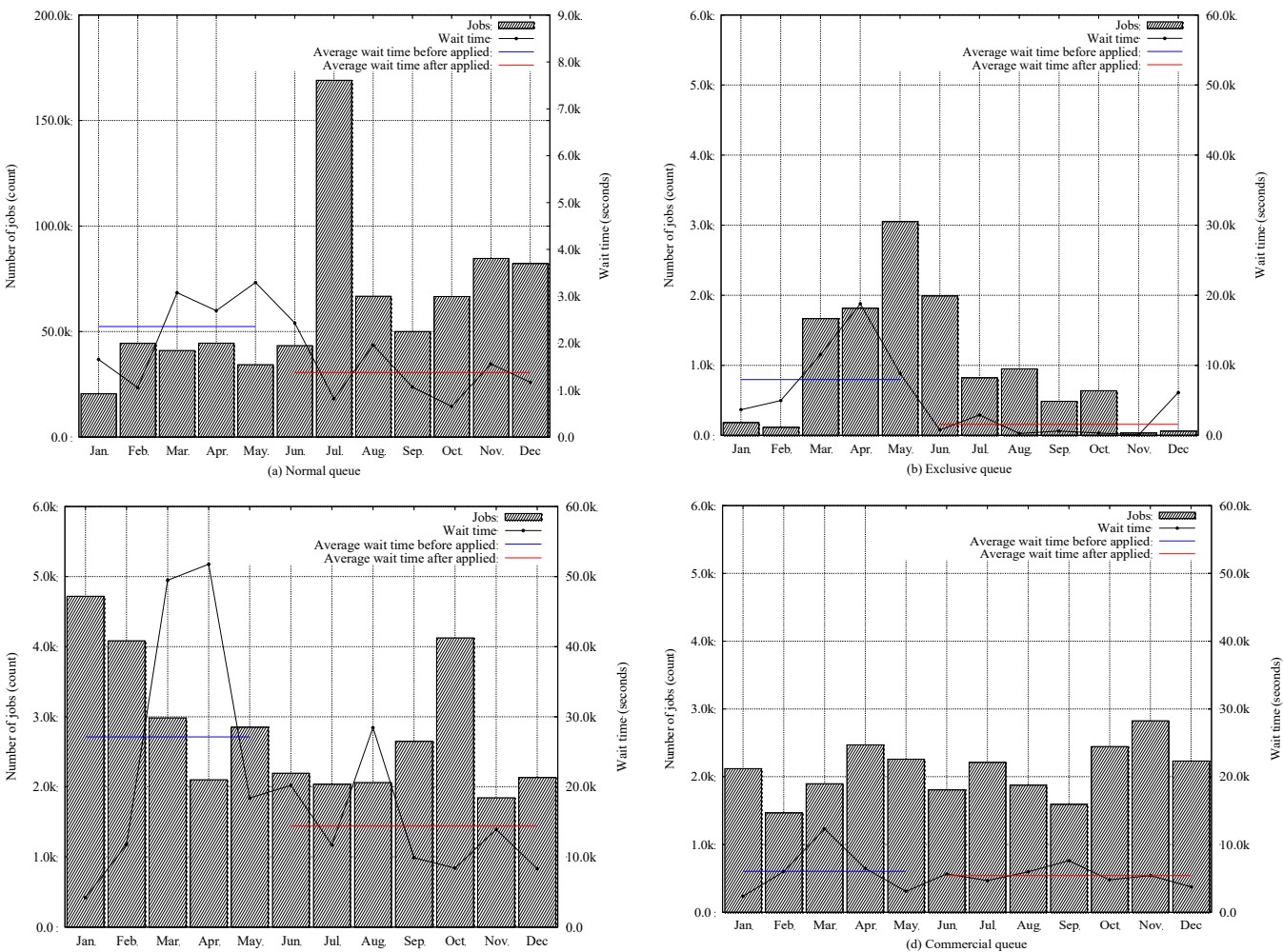

**Figure 6.** Comparison of the waiting time before and after applying K-hook with scheduling by main queues. (**a**) Normal queue and (**b**) exclusive queue in KNL type. (**c**) Norm_skl queue and (**d**) commercial queue in SKL type.

These results show that from the user's perspective, the waiting time does not increase because of our method being applied to each queue. Although it cannot be generalized due to the unique characteristics of the user's jobs, this is an attempt to satisfy the job success rate from the operator's perspective and ensure the user's convenience. As this is a real service supercomputing environment, we could not compare the before and after scenarios perfectly. Because the job submitted status of users may be different every month, some months may have numerous jobs, and some may have fewer. Therefore, we analyzed the waiting time as a rough trend by displaying the monthly average for one year on a graph.

### 5.3. MTBI of Supercomputing Service

The MTBI, which represents the average time between interrupts (more specifically, the average time for which the system operates continuously without interruption) and

MTBF, which represents the average time between failures of the system, are usually used as indicators of system reliability. The MTBI was calculated using Equation (1).

$$MTBI_{service} = \frac{production\ time}{number\ of\ service\ interrupts} \tag{1}$$

In 2019, regular monthly maintenance was conducted on the Nurion system to reduce failures. Despite this, three service interruptions occurred in June and July because of parallel file system failures. Table 3 summarizes the number of interrupts for each month due to system failure, including the periodic downtime in 2019. The table shows that the Nurion system experienced 15 interrupts in 2019, and its MTBI was 24.3 days (=365/15).

**Table 3.** The number of system interruptions for maintenance.

| Month | Jan | Feb | Mar | Apr | May | Jun | Jul | Aug | Sep | Oct | Nov | Dec |
|---|---|---|---|---|---|---|---|---|---|---|---|---|
| Interrupts | 1 | 1 | 1 | 1 | 1 | 3 | 2 | 1 | 1 | 1 | 1 | 15 |

This indirectly shows that the Nurion system operated in a remarkably stable manner during the first year of use. Figure 7 shows the monthly system utilization and service availability rates. System utilization is the ratio of the total CPU time consumed by users' jobs to the total CPU time of all available nodes. It is the ratio of the production time (operation time of all 8437 nodes minus planned maintenance time minus failure time) to total time. The system availability rate is the fraction of a period in which an item is in a condition to perform its intended function upon demand. This was calculated using Equation (2). The bar graph in Figure 7 shows that the system utilization gradually increased. In addition, system availability was maintained at an average of 97% or more.

$$Availability_{system}\ (\%) = \frac{uptime}{total\ time} \times 100 \tag{2}$$

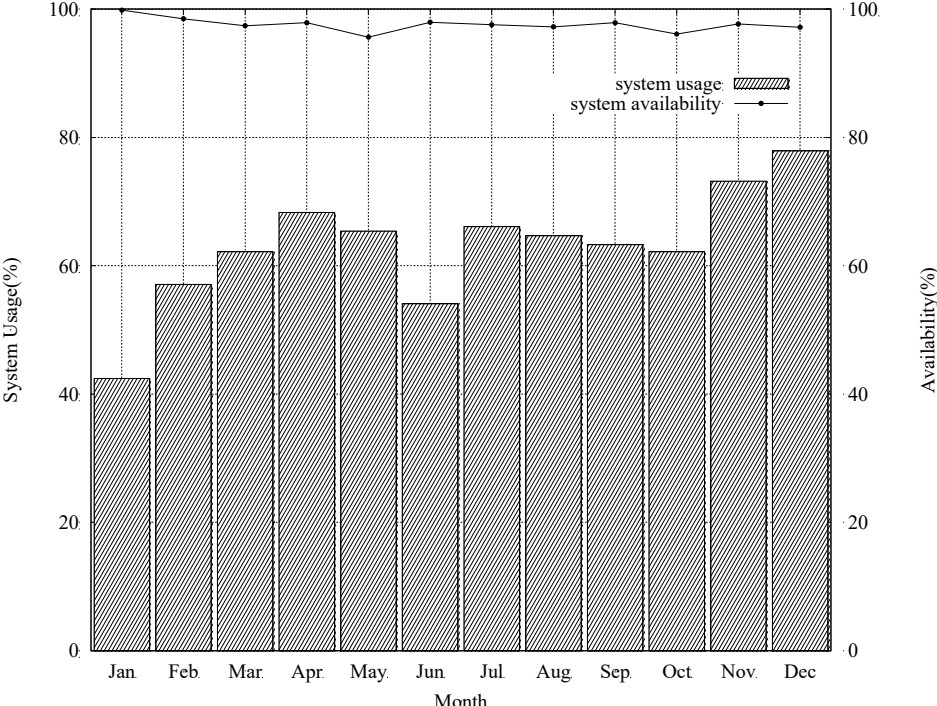

**Figure 7.** Utilization (bar graph, left *y*-axis) and availability (line graph, right *y*-axis) rates of the Nurion system in 2019.

## 6. Conclusions

Designing job scheduling in HPC environments for supercomputing services is highly complex. In particular, scheduling for large-scale resources emphasizes efficient operation and the improvement of service availability. We analyzed the frequent occurrence of job submission errors based on the operational data of supercomputer Nurion for several months. By analyzing these error codes in detail, a method named K-hook was developed to increase the job-submission success rate and was added to the scheduling architecture. Consequent to applying the K-hook method to scheduling, the job-submission success rate was improved by 15% without negatively affecting the user's waiting time. The method proposed in this paper is being applied in an environment that provides actual services and is expected to provide both quality service and valuable research insights. In the future, we will continue exploring innovations that apply intelligent and automated technologies through a deeper analysis of user data generated from supercomputer operations to reduce users' waiting time and improve service availability.

**Author Contributions:** Conceptualization, J.-K.L. and M.-W.K.; methodology, D.-S.A.; software, M.-W.K. and D.-S.A.; validation, J.Y. and J.W.; formal analysis, J.-K.L.; investigation, S.-J.K.; resources, J.W. and T.H.; data curation, G.L.; writing—original draft preparation, J.-K.L.; writing—review and editing, G.L.; visualization, M.-W.K.; J.Y.; project administration, T.H.; funding acquisition, T.H. All authors have read and agreed to the published version of the manuscript.

**Funding:** This research has been performed as a project of Project No. K-21-L02-C01-S01 (The national flagship supercomputer infrastructure construction and service) supported by the KOREA INSTITUTE of SCIENCE and TECHNOLOGY INFORMATION (KISTI).

**Institutional Review Board Statement:** Not applicable.

**Informed Consent Statement:** Not applicable.

**Data Availability Statement:** No new data were created or analyzed in this study. Data sharing is not applicable to this article.

**Acknowledgments:** This study has been performed as a sub-project of KISTI's project "The national flagship supercomputer infrastructure construction and service".

**Conflicts of Interest:** The authors declare no conflict of interest.

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
