# Peer review of "Improvements to Supercomputing Service Availability Based on Data Analysis"

_applsci, doi:10.3390/app11136166_

Round 1
Reviewer 1 Report
The authors present a method named K-hook to increase the job-submission success rate in an HPC environment. The study was conducted over several months and has details of why jobs fail. By applying the K-Hook scheduling (essentially error checking before and after job execution) the authors show that the job-submission success rate was improved by 15 percentage points without negatively affecting the user’s waiting time. The study was conducted on an actual supercomputer queue, making it valuable for other such environments.
The paper reads well and the scientific method is well thought out. Table 2 is interesting and shows why jobs fail most of the time. Job success rate before and after applying K-Hook is pronounced.
There are several minor problems that authors should address. The authors do not explain why the improvements happen (Section 5 - K-Hook catches a combination of errors, which ones are those?) Figure 6 also does not explain why improvements happen. Similarly, why are the improvements different for different queues? (10% for commercial queues vs 34% in normal queues). Are they artifacts of difference in job types? If so, how does it affect the overall efficiency of the K-Hook algorithm (figure 5)? Maybe the efficiency increased because more normal jobs were submitted after June than commercial jobs?
The other question is why are the checks performed in MoM module and not in the scheduler phase? Performing checks such as "no such file and directory" might be more efficient in the scheduling phase. The authors do not explain why.
Finally, the authors do not explain what happens when a job fails. Is it sent back to the user with a feedback of why it failed? This can be useful but the authors do not mention this.
To summarize, the authors present a practical analysis of scheduling jobs on a supercomputer. They show that their proposed method can improve success rate by 15%. However, the results and the underlying reasons behind them need to be better explained.
Author Response
Comment 1: The authors present a method named K-hook to increase the job-submission success rate in an HPC environment. The study was conducted over several months and has details of why jobs fail. By applying the K-Hook scheduling (essentially error checking before and after job execution) the authors show that the job-submission success rate was improved by 15 percentage points without negatively affecting the user’s waiting time. The study was conducted on an actual supercomputer queue, making it valuable for other such environments.
The paper reads well and the scientific method is well thought out. Table 2 is interesting and shows why jobs fail most of the time. Job success rate before and after applying K-Hook is pronounced.
Response: First of all, thank you for acknowledging the value of our paper. Your review comments helped us to improve the paper.
Comment 2: There are several minor problems that authors should address. The authors do not explain why the improvements happen (Section 5 - K-Hook catches a combination of errors, which ones are those?) Figure 6 also does not explain why improvements happen. Similarly, why are the improvements different for different queues? (10% for commercial queues vs 34% in normal queues). Are they artifacts of difference in job types? If so, how does it affect the overall efficiency of the K-Hook algorithm (figure 5)? Maybe the efficiency increased because more normal jobs were submitted after June than commercial jobs?
Response: We have added the following explanations as to why the job success rate improved in Figure 5.
- In section 3, according to Table 2, we added a detailed analysis of why the job failed as well as a description of how it matches the proposed K-hook functions (lines 133-150).
- In subsection 2, we mentioned the matching points between error messages and K-hook functions (lines 204-205, 208-209, 212-214, 224-226).
- In subsection 5.1, we briefly explained why the improvements occurred (lines 252-255).
Regarding Figure 6, we explained why we analyzed waiting time by queues. The K-hook we developed does not directly reduce a user’s waiting time. Rather, in principle, the job request passes through the K-hook and goes through various checks. If it does not pass, this node is processed offline and set to the scheduler nodes. Thus, the scheduler needs to wait longer to be assigned a compute node again, thereby increasing the waiting time. We modified each queue policy to reduce waiting time. These explanations have been added as follows:
- In subsection 4.1, we explained the reason for the longer waiting time when adding K-hooks (lines 178-183).
- In subsection 5.2, we explain how we reduced a user’s waiting time and the limitations indicated by the data in Figure 6 (lines 259-264; lines 283-288).
Comment 3: The other question is why are the checks performed in MoM module and not in the scheduler phase? Performing checks such as "no such file and directory" might be more efficient in the scheduling phase. The authors do not explain why.
Response: We implemented K-hook to MoM rather than Scheduler to reduce the load of scheduler nodes, which handle thousands of compute nodes in a huge HPC environment. Because our supercomputer scheduled over 8,000 compute nodes, we added K-hook to MoM on compute nodes. When a job is submitted, it is checked with the functions defined through K-hook, and if it does not pass, the node is processed offline and Server is notified through Communication. Performing checks such as “no such file and directory” occurs when the scheduler has already allocated compute node and executed user programs on it. Hence, it is reasonable to implement K-hook to MoM on compute nodes (lines 168-177).
Comment 4: Finally, the authors do not explain what happens when a job fails. Is it sent back to the user with a feedback of why it failed? This can be useful but the authors do not mention this.
Response: When a user’s job fails, the scheduler sends a failure message to the user through the mail. This message contains a simple summary; for more detailed error messages, the *.log and *.err files are referred to in the user’s home directory (lines 119-122).
Comment 5: To summarize, the authors present a practical analysis of scheduling jobs on a supercomputer. They show that their proposed method can improve success rate by 15%. However, the results and the underlying reasons behind them need to be better explained.
Response: The results and the underlying reasons behind them have been further explained based on the above questions. We also added a summary to subsection 5.1 (lines 252-255).
Reviewer 2 Report
The paper presented a methodology for improving the availability of supercomputing resources by reducing the number of failing jobs. Basically, the authors proposed to detect (possibly) preventively the conditions that lead a job to fail; by detecting potential failing jobs should allow to reschedule them along with the others in a more effective and fair way. The paper and the authors claims are well supported by experimental data. Results of tests done with a production supercomputer demonstrated to be effective.
There are few minor things in my opinion the authors should improve in the text. One is explaining better how the improved workflow (i.e., the way a job is submitted, the application of hooks and the way a potential failing condition detection is used to reschedule the job) helps the HPC system administrators to make rescheduling decisions of the potential failing jobs. The second one is the lack of some detail regarding the way hooks (job checking functions) have been implemented (are they pieces of Python, C++, etc. code? are they bash scripts?) which would help to better understand the way they are integrated in the batch scheduling system. In this regard, is the proposed methodology extensible to other batch schedulers other than PBS pro?
Author Response
Comment 1: The paper presented a methodology for improving the availability of supercomputing resources by reducing the number of failing jobs. Basically, the authors proposed to detect (possibly) preventively the conditions that lead a job to fail; by detecting potential failing jobs should allow to reschedule them along with the others in a more effective and fair way. The paper and the authors claims are well supported by experimental data. Results of tests done with a production supercomputer demonstrated to be effective.
Response: Thank you for your positive comments.
Comment 2: There are few minor things in my opinion the authors should improve in the text.
One is explaining better how the improved workflow (i.e., the way a job is submitted, the application of hooks and the way a potential failing condition detection is used to reschedule the job) helps the HPC system administrators to make rescheduling decisions of the potential failing jobs.
Response: We have added two parts to the revised manuscript in response to your comment.
- In section 3, we have described what will happen when a job fails. (lines 119-122)
- In subsection 4.1, we have described what will happen when a job is submitted through K-hook. Precisely, when a job is submitted, it is checked with the functions defined through K-hook, and if it does not pass, the node is processed offline and the server is notified. (lines 172-177)
Comment 3: The second one is the lack of some detail regarding the way hooks (job checking functions) have been implemented (are they pieces of Python, C++, etc. code? are they bash scripts?) which would help to better understand the way they are integrated in the batch scheduling system. In this regard, is the proposed methodology extensible to other batch schedulers other than PBS pro?
Response: K-hook uses Python plugins provided by PBSPro hook. The main functions of K-hook are implemented in a bash script, which is called through PBSPro hooks in Python. The bash script can be extensible to any other batch schedulers (lines 184-186).